# Technological Characterization of Lactic Acid Bacteria Strains for Potential Use in Cheese Manufacture

**DOI:** 10.3390/foods12061154

**Published:** 2023-03-09

**Authors:** Fabrizio Domenico Nicosia, Alessandra Pino, Guilherme Lembi Ramalho Maciel, Rosamaria Roberta Sanfilippo, Cinzia Caggia, Antonio Fernandes de Carvalho, Cinzia Lucia Randazzo

**Affiliations:** 1Department of Agriculture, Food and Environment, University of Catania, 95123 Catania, Italy; fabrizio.nicosia@phd.unict.it (F.D.N.); alessandra.pino@unict.it (A.P.);; 2ProBioEtna SRL, Spin off of the University of Catania, Via Santa Sofia, 100, 95123 Catania, Italy; 3CERNUT, Interdepartmental Research Centre in Nutraceuticals and Health Products University of Catania, 95125 Catania, Italy; 4InovaLeite—Laboratório de Pesquisa em Leite e Derivados, Departamento de Tecnologia de Alimentos, Universidade Federal de Viçosa, Viçosa 36570900, MG, Brazil

**Keywords:** adjunct culture, lactic acid bacteria, aminopeptidases, diacetyl production, exopolysaccharide production, PepX, Pep N

## Abstract

A total of 26 lactic acid bacteria isolates from both Italian and Brazilian cheeses were tested for their use in cheesemaking. Isolates were screened for salt tolerance, exopolysaccharide and diacetyl production, lipolytic, acidifying, and proteolytic activities. In addition, the aminopeptidase (Pep N and Pep X) activities, were evaluated. Most of the strains demonstrated salt tolerance to 6% of NaCl, while only two *L. delbruekii* (P14, P38), one *L. rhamnosus* (P50) and *one L. plantarum* (Q3C4) were able to grow in the presence of 10% (*w*/*v*) of NaCl. Except for 2 *L. plantarum* (Q1C6 and Q3C4), all strains showed low or medium acidifying activity and good proteolytic features. Furthermore, lipolytic activity was revealed in none of the strains, while the production of EPS and diacetyl was widespread and variable among the tested strains. Finally, regarding aminopeptidase activities, 1 *L. delbrueckii* (P10), 1 *L. rhamnosus* (P50), and 1 *L. lactis* (Q5C6) were considered as the better performing, showing high values of both Pep N and Pep X. Based on data presented here, the aforementioned strains could be suggested as promising adjunct cultures in cheesemaking.

## 1. Introduction

Lactic acid bacteria (LAB), normally present in milk, utensils, and surfaces of dairy farms, play an important role in cheesemaking, both in the early stages of milk fermentation and during cheese ripening [1,2]. In particular, the starter LAB (SLAB), rapidly fermenting the lactose, produces high concentrations of lactic acid whereas the non-starter LAB (NSLAB) is mainly involved during the ripening process in the definition of the sensory profile of the final product [3]. The NSLAB carefully selected based on particular metabolic features and intentionally added as “adjunct cultures”, can improve the taste, flavor, and texture of cheese [4]. Three primary LAB metabolic pathways, including lactate and free fatty acid release metabolisms as well as proteolysis (with subsequent peptides and amino acid catabolism), are mainly involved in the definition of the sensory profile of the final product [5]. In addition, by fermenting citrate, glucose, lactose, and other carbon sources, the produced diacetyl and acetoin confers desirable sensory features to ripened cheeses [6,7]. In particular, diacetyl plays a key role in the flavor development of Dutch-style cheeses, cottage cheese, quark, and many other fermented dairy products [8]. Moreover, different studies demonstrated that some LAB provides an important contribution to the development of the texture of cheeses through, for example, the production of exopolysaccharides (EPS) [9,10]. As stabilizers, EPS can improve the firmness of the casein network by binding water and interacting with other milk components, such as whey proteins and casein micelles, along with promoting antimicrobial and antioxidant activities in dairy products [11]. Furthermore, LAB possesses a complex enzymatic system that includes proteinases and peptidases. Peptidases are able to hydrolyze small peptides starting from long peptide chains, which in turn are freed from milk caseins by the action of coagulating enzymes and microbial proteinases themselves [12]. The accumulation of peptides with the presence of hydrophobic amino acids (lysine, leucine, and proline), in the peptide side chains, is generally associated with the development of a bitter taste in cheese [13]. Some enzymes, produced by LAB, are able to selectively hydrolyze these peptides reducing the bitter taste. From this perspective, aminopeptidase N (Pep N) is one of the most significant peptidases able to hydrolyze peptides with hydrophobic N-terminal amino acids (e.g., Leu and Lys). Moreover, X-prolyl-dipeptidyl aminopeptidase (PepX) is an additional peptidase with the ability to hydrolyze proline in specific peptide sites mitigating the bitter taste of the cheese [14]. All these properties are important for the selection of an adjunct culture able to improve both the taste and texture of cheeses. All the technological properties mentioned above contribute to flavor development and quality of the cheese, which are strongly determined by the complex dynamics and interaction among cheese indigenous LAB [15]. In particular, the indigenous LAB population arising from milk and dairy environment, usually consisting of streptococci, enterococci, and lactobacilli, is considered a source of enzymatic activity involved in flavor formation, especially in traditional cheeses [15].

The aim of this work was to characterize the technological features of LAB strains, isolated from different traditional cheeses, in order to set up an adjunct culture for cheesemaking. In particular, indigenous strains were isolated from Pecorino Siciliano and from Ragusano cheeses which are traditional Italian hard cheeses manufactured from raw ewe’s and cow’s milk, respectively, without any starter cultures, using ancient manufacturing processes and ripened at least for 4–6 months or longer [2,16]. Brazilian strains were isolated from Marajó cheese, a traditional Brazilian cheese produced in the Marajó island, made from raw buffalo milk, in accordance with the region’s cultural traditions by spontaneous coagulation, and characterized by a complex flavor mainly generated by autochthonous LAB. This cheese has been produced for more than a century and is the socioeconomic foundation of tiny farmers on the island [17]. All strains were tested for the ability to exert lipolytic and proteolytic activities, produce exopolysaccharides and diacetyl, and tolerate different NaCl concentrations. In addition, the presence of specific aminopeptidases (Pep N and Pep X), able to hydrolyze bitter peptides, was in-depth evaluated.

## 2. Materials and Methods

### 2.1. Bacterial Strains and Culture Conditions

A total of 26 LAB strains were used in this work: 11 belonging to the collection of InovaLeite (Laboratory of Milk and Dairy Products, Universidade Federal de Viçosa, Brazil) were isolated from Marajó and Pará cheeses using M17 agar medium (Difco laboratories, Ditroit, MI, USA) and MRS agar medium (Difco laboratories, Ditroit, MI, USA), and 15, belonging to the collection of the Laboratory of Microbiology (Department of Agricultural, Food, and Environment, Di3A, University of Catania, Italy), were isolated from Pecorino and Ragusano cheeses using Rogosa agar medium (Oxoid Ltd., Basingstoke, UK) and MRS agar medium (Oxoid Ltd., Basingstoke, UK). The strains were previously identified at species level through molecular methods as follows: 4 *Lactiplantibacillus plantarum* (Q1C6, Q3C4, Q5C9, and Q6C4), 6 *Pediococcus acidilactici* (Q1C8, Q3C1, Q3C3, Q6C1, Q6C5, and Q22C2), 1 *Lactococcus lactis* subsp. *lactis* (Q5C6), 14 *Lactobacillus delbrueckii* (P7, P9, P10, P11, P12, P13, P14, P15, P33, P36, P37, P38, P39, and P40) and 1 *Lacticaseibacillus rhamnosus* (P50). The strains were stored in de Man Rogosa and Sharpe (MRS) broth (Oxoid Ltd., Basingstoke, UK) with 20% (*v*/*v*) glycerol and kept at −20 °C till further use.

### 2.2. Bacterial Growth and Cell Suspension Standardization

LAB isolates were cultured in MRS broth (Oxoid Ltd., Basingstoke, England) and incubated at 30 °C for 24 h. After growth, cells were centrifuged at 12,000 rpm for 2 min at 4 °C and the obtained pellet was suspended in NaCl (0.9% *w*/*v*) to a turbidity equivalent to the 1 McFarland (about 3 × 10^8^ CFU/mL). The standardized cell suspension was used to test proteolytic, lipolytic, and acidifying activities, diacetyl and exopolysaccharides production as well as salt tolerance.

### 2.3. Proteolytic and Lipolytic Activities

The extracellular proteolytic activity was evaluated following the protocol described by Randazzo and co-workers (2021) [18] and Franciosi and co-workers (2009) [19]. In detail, 2 µL of cells, standardized as previously described, were spotted on the surface of petri dishes containing Plate Count Agar (PCA, HiMedia, Mumbai, MH, India) supplemented with skim milk (10% *w*/*v*, Oxoid Ltd., Basingstoke, UK). After incubation at 30 °C for four days, plates were checked for the presence of a clear zone surrounding the colonies.

The lipolytic activity was tested by spotting 2 µL of standardized cells (about 3 × 10^8^ CFU/mL) on the lipolytic medium composed of peptone, 2.5 g/L; casein peptone, 2.5 g/L; yeast extract, 3 g/L; Agar, 12 g/L; and 1% of tributyrin. After incubation at 30 °C for 72 h, the presence of a halo around the bacterial spot revealed lipolytic activity [20].

Each test was performed in triplicate.

### 2.4. Diacetyl and Exopolysaccharide Production

In order to measure the diacetyl production, 0.1 mL aliquots of standardized cell suspension were added to 10 mL of sterile reconstituted skim milk (10% *w*/*v*, Oxoid Ltd., Basingstoke, UK), and the mixture was incubated at 30 °C for 24 h. Then, an aliquot of 1 mL was transferred to a sterile tube and supplemented with 0.5 mL of α-naphthol (1% *w*/*v*) and KOH (16% *w*/*v*), then the mixture was incubated at 30 °C for 10 min. Diacetyl production was indicated by the formation of a pink ring and classified as weak (+), medium (++), or strong (+++) according to the color intensity.

The feature relative to the synthesis of exopolysaccharide (EPS) was assessed using the methodology described by Dal Bello et al. (2012) [21]. In detail, 0.1 mL aliquot of the standardized cell suspension was inoculated into 10 mL of sterile reconstituted skim milk (10% *v*/*v*, Oxoid Ltd., Basingstoke, UK) and the mixture was incubated at 30 °C for 48 h. The presence of stringiness was used to evaluate the EPS production.

Each test was performed in triplicate.

### 2.5. Salt Tolerance

The ability of the tested isolates to grow at different salt concentrations was evaluated as described by Ferrari et al. (2016) [22]. MRS broth medium (Oxoid, Milan, Italy) containing bromocresol purple (0.04% *w*/*v*) and different NaCl concentrations (2, 6, 10% *w*/*v*) was transferred into sterile tube and inoculated with 1% of standardized cell suspension. After incubation at 37 °C for one week, the change of color from purple to yellow was considered as positive growth.

The test was performed in triplicate.

### 2.6. Acidifying Activity

Acidifying activity was determined by inoculating the standardized cell suspension (2% *v*/*v*) into reconstituted skim milk (10% *w*/*v*, Oxoid Ltd., Basingstoke, UK). The pH changes (∆pH) were determined after 6 h and 8 h of incubation at 37 °C using a pH meter (MettlerDL25, Mettler-Toledo International Inc., Columbus, OH, USA).

The test was performed in triplicate.

### 2.7. Aminopeptidase Activity

#### 2.7.1. Cell Free Extract Preparation

LAB strains were grown at 37 °C in MRS broth medium (Oxoid, Milan, Italy) until reaching the late exponential phase. After centrifugation (10.000 *g*, 10 min, 4 °C), cells were washed twice with sodium phosphate buffer (0.05 M, pH 7.0), then standardized to 10^9^ CFU/mL in the same buffer (0.05 M, pH 7.0). To obtain the cell free extract (CFE), cell lysis was carried out through the bead-beating method, in the Precellys apparatus (Bertin technologies, Düsseldorf, Germany), using zirconium beads with a diameter of 0.1 mm. The treatment at 6000 rpm for 40 s was repeated twice. Samples were placed on ice (for 4 min) between each cycle. At the end of the treatment, the samples were placed on ice for 10 min. CFE was obtained after removing zirconium beads, cell debris, and unbroken cells by centrifugation (10.000 *g*, 10 min, 4 °C). The protein concentration was determined by using the Pierce™ BCA Protein Assay Kit (Thermo Fisher, Waltham, MA, USA).

#### 2.7.2. Aminopeptidase N Activity

Aminopeptidase activity of the CFE was determined as described by Requena et al. (1993) [23]. Then, 100 μL of CFE were added to 80 μL of phosphate buffer (50 mM, pH 7.0) and 20 μL of a reaction buffer containing the substrate Lys-p-nitroanilide dihydrobromide (20 mM in methanol) (Sigma-Aldrich, St. Louis, MO, USA). After incubation at 37 °C for 30 min (or until the mixture reaches a strong yellow color) the reaction was stopped by adding 500 μL of glacial acetic acid (10% *v*/*v*, Panreac, Barcelona, Spain). Optical density (OD) at 410 nm was measured using iMark™ Microplate Absorbance Reader (Biorad, Milan, Italy). The test was performed in triplicate, using the “white test” (20 μL of the reaction buffer containing the substrate Lys-p-nitroanilide dihydrobromide with 180 μL of phosphate buffer) as blank. The aminopeptidase N (Pep N) activity was expressed as U/mg of protein. One U was defined as the amount of enzyme required to release 1 μmol of p-NA per minute under the assay conditions.

#### 2.7.3. Aminopeptidase X Activity

For the evaluation of aminopeptidase X (Pep X), 50 μL of CFE were added to 600 μL of phosphate buffer (50 mM, pH 7.2) and 50 μL of a reaction buffer containing the substrate H-Ala-Pro-p-nitroanilide HCl (20 mM in methanol) (ChemCruz Biochemicals, Santa Cruz, CA, USA). After incubation at 37 °C for 30 min (or until the mixture reaches a strong yellow color), the reaction was stopped by adding 500 μL of glacial acetic acid (10% *v*/*v*, Panreac, Barcelona, Spain). Optical density (OD) at 410 nm was measured using iMark™ Microplate Absorbance Reader (Biorad, Milan, Italy). The test was performed in triplicate, using the “white test” (50 μL of a reaction buffer containing the substrate H-Ala-Pro-p-nitroanilide HCl with 150 μL of phosphate buffer) as blank. Specific activity was expressed as U/mg of protein. One U was defined as the amount of enzyme required to release 1 μmol of p-NA per minute under the assay conditions.

### 2.8. Statistical Analysis

Pep N and Pep X data were subjected to One-way ANOVA analysis with Tukey’s post hoc test using the Statistica software (TIBCO Software, Palo Alto, CA, USA). Differences were considered statistically significant at *p* < 0.05. In order to correlate acidifying and aminopeptidase activities, data were subjected to principal component analysis (PCA) using the XLSTAT (2023.1.1.1397) software.

## 3. Results and Discussion

### 3.1. Proteolytic and Lipolytic Activities, Diacetyl and Exopolysaccharide Production

Results of proteolytic and lipolytic activities as well as diacetyl and exopolysaccharide production are shown in Table 1. Overall, all the tested strains, with the exception of two *L. plantarum* strains (Q1C6 and Q3C4), showed proteolytic activity. It is well known that LAB possesses an efficient proteolytic system, with complex combinations of proteinases and peptidases, which allow them to obtain organic nitrogen from complex proteins like casein [24]. Proteases hydrolyze caseins forming peptides which, crossing the cell membrane through specific transport proteins, are further degraded into amino acids by intracellular peptidases [25]. The amino acids, through specific catabolic pathways, are transformed into volatile and non-volatile compounds, which play a key role in the definition of the sensory properties of cheese [26].

Concerning lipolytic activity, as reported in Table 1, none of the strains exhibited this feature. Identical results were reported by Meng et al. (2018) [27] and Monfredini et al. (2012) [28]. In fact, none of the tested LAB strains showed lipolytic activity through tributyrin agar, while Silva et al. (2020) [29] found lipolytic activity only in 5 strains of LAB out of 37 tested. The inability of LAB strains to break down milk fat during ripening makes them suitable as adjunct cultures. In fact, it is well known that NSLAB should not possess lipolytic activity since this is associated with the development of rancid flavor [30,31,32]. Differently, one important feature of adjunct cultures is related to diacetyl production. Out of the 26 strains, 20 showed diacetyl production (Table 1); the strains isolated from Pecorino cheese were classified as medium diacetyl producers whereas LAB strains from Ragusano, Marajó, and Pará cheeses showed variable ability to produce diacetyl (Table 1). In detail, 4 *L. delbrueckii* (P11, P12, P13, and P39) and one *P. acidilactici* (Q3C1) strains displayed strong production, 4 *L. delbrueckii* (P7, P9, P10, and P36) 2 *L. plantarum* (Q5C9 and Q6C4), and one *L. lactis* (Q5C6) had medium production, 3 *P. acidilactici* (Q1C8, Q6C5 and Q22C2), 2 *L. plantarum* (Q1C6 and Q3C4), one *L. delbrueckii* (P38), and one *L. rhamnosus* (P50) strains showed weak production, while 4 *L. delbrueckii* (P14, P15, P33, and P40) and 2 *P. acidilactici* (Q3C3 and Q6C1) strains had no production. This wide variability in diacetyl production was previously observed in *L. rhamnosus* strains isolated from semihard goat cheese [27], and in *L. rhamnosus* and *L. dellbrueckii* subsp. *bulgaricus* strains isolated from hard raw cow’s milk cheese [28]. This evidence is in accordance with previously reported data, suggesting that diacetyl production is a strain-dependent feature [19,33]. The production of this compound represents a very important variable in the choice of a strain for the formulation of an adjunct culture; in fact, the diacetyl, produced by LAB using citrate, glucose, lactose, and other carbon sources as substrates, confers a buttery taste to dairy products [34,35]. Furthermore, diacetyl was shown to have inhibitory activity against foodborne pathogens, especially when combined with bacteriocin such as nisin [36].

Concerning EPS production, three *L. delbrueckii* strains (P10, P14, and P38) isolated from Pecorino and Ragusano cheeses showed the ability to produce these compounds (Table 1). The results are in agreement with those previously reported by Christianah et al. (2008) [37] and Khubaib et al. (2018) [38], showing that strains belonging to *L. delbrueckii* and *L. delbrueckii* subps. *bulgaricus* were able to produce EPS and among them, *L. delbrueckii* subsp. *bulgaricus* is a well-known EPS producer [39]. In detail, *L. delbrueckii* subps. *bulgaricus* exhibit many intraspecific biosynthetic pathways producing different EPS structures, consisting of units of repeated monomers such as glucose, galactose, rhamnose, and sometimes fructose [40]. *L. delbrueckii* subps. *bulgaricus* EPS producers have been used for many years in classical yogurt, drinking yogurt, fresh cheeses, cultured cream, or milk-based desserts, thanks to their ability to increase viscosity, prevent syneresis and improve sensory and nutritional characteristics of dairy products [41,42]. In the work of Ahmed et al. (2005) [43], the employment of an EPS-producing culture, during the manufacture of the Karish cheese, strongly influenced the textural properties of the final product. The resulting cheese exhibited lower hardness, consistency, chewiness, and adhesiveness compared to the cheese obtained using a strain unable to produce EPS. Furthermore, the panelists described the cheese made with the EPS-producing strain as smooth, creamy, moist, and soft, while the one with the EPS-nonproducing strain was described as dry and granular. In addition, several functional features were recently attributed to EPS produced by LAB, such as antitumorigenic, antimicrobial, and antioxidant activities. In particular, EPS are able to counteract reactive oxygen species (ROS) hindering the development of many disorders including lung injury, atherosclerosis, inflammation, aging, and cancer [44].

### 3.2. Salt Tolerance and Acidifying Activity

As reported in Table 2, all tested strains were able to grow in presence of 2% (*w*/*v*) of NaCl. Moreover, 13 strains (7 *L. delbruekii*, 2 *L. plantarum*, 3 *P. acidilactici* and one *L. lactis*) demonstrated salt tolerance at the concentration of 6% (*w*/*v*), and a total of 4 strains (2 *L. delbruekii* (P14 and P38), 1 *L. rhamnosus* (P50) and 1 *L. plantarum* (Q3C4) grew in presence of 10% (*w*/*v*) of NaCl. The results agree with the data reported by Meng et al. (2018) [27], which demonstrated the ability of *L. rhamnosus* strains isolated from semihard goat cheese to grow in presence of high NaCl concentration (10% *w*/*v*). In addition, *L. rhamnosus* isolates from traditional Provola dei Nebrodi cheese showed high tolerance to the presence of NaCl (10% *w*/*v*) resulting in accordance with evidence reported in the present study [18]. Differently, the salt resistance feature is not widespread in *L. plantarum* strains; in fact, our data revealed that only 1 strain out of 4 *L. plantarum* strains demonstrated 10% (*w*/*v*) NaCl tolerance, confirming data reported by Karasu et al. (2010) [45] where only 2 out of 12 *L. plantarum* strains were able to grow at a salt concentration of 9% (*w*/*v*), suggesting the low tolerance of *L. plantarum* strains to high salt concentration. To tolerate high salt concentrations, LAB develops various strategies such as the uptake or the synthesis of a limited number of solutes [46]. The ability of NSLAB strains to grow in a wide range of salt concentrations is fundamental since they are often subjected to high concentrations of NaCl especially during brining and ripening of cheese production. In fact, NaCl is a common preservative for long-term storage cheeses and is crucial for managing cheese ripening [47].

Regarding the acidifying parameter generally, to be classified as a starter, a bacterial strain should be able to lower the pH of the milk to 5.3 after 6 h [48]. According to that, all strains are often classified into three groups in line with their rate of acidification: fast acidification (pH ˂ 5.3 after 6 h of fermentation), medium (pH < 5.3 after 8 h of fermentation), and slow (pH > 5.3 after 8 h of fermentation). As reported in Figure 1, the strains tested in the present study showed a ∆pH within 6 h ranging from 0.16 to 2.01, while the ∆pH within 8 h was from 0.22 to 2.08. According to that and as displayed by PCA plot (Appendix A), 2 *L. plantarum* strains (Q1C6 and Q3C4) demonstrated fast acidifying activity, 5 *L. delbruekii* stains (P14, P15, P36, P37, and P38) medium behavior, while all the others showed low acidifying activity proving to be usable for an adjunct culture. Similar results were obtained by Hadef et al., (2022) [49] demonstrating the weak acidification among the 36% LAB strains tested. Moreover, lactobacilli strains isolated from Tenerife cheese by Pérez et al. (2003) [50] revealed low acidification activity. It is well known that LAB strains, selected as adjunct cultures, should have low acidifying activity; in fact, a high rate could generate sensory defects in cheese [51].

### 3.3. Aminopeptidase Activities

In the present study, the aminopeptidase activities Pep N and Pep X were tested using Lys-pNa and H-Ala-Pro-pNa substrates, respectively. In detail, as reported in Figure 2, the tested strains showed high activity against the substrate Lys-pNa, with values ranging from 265.47 to 2.15, of units per milligram of protein per minute (U/mg). In particular, according to the PCA plot (Appendix A), the strain Q5C6 ascribed to the *L. lactis* species, as well as the *L. delbrueckii* P10 strain and the *L. rhamnosus* P50 strain were considered the best performers. Pep N data present in this study were higher to those obtained by Morea et al., (2007) [52] in fact, *L. delbrueckii*, *L. gasseri* and *P. pentosaceus* strains isolated from Caciocavallo Pugliese cheese showed values ranging from 45.3 to 10.01 U/mg. Further, González and co-workers (2010) [53] demonstrated low Pep N activity among LAB strains isolated from traditional Spanish cheese; in fact, except for few cases, the Pep N activity was generally low or even undetectable. Moreover, the *L. rhamnosus* P50 strain, tested in the present study, showed very high activity against the Lys-pNa substrate (254.56 U/mg) compared to *L. rhamnsosus* strain tested by Carafa et al. (2015) [31], which revealed an activity equal to 19 U/mg.

Overall, it is well known that aminopeptidase activity is crucial for the breakdown of peptides and the release of amino acids during the secondary proteolysis of the cheese. In particular, Pep N displays high selectivity for the basic amino acids Lys and Arg, followed by the hydrophobic/uncharged residues Leu and Ala [54]. It was previously isolated from strains ascribed to different species such as *Lacticaseibacillus casei*, *Lactobacillus delbrueckii*, *Lactobacillus helveticus,* and *Streptococcus thermophilus* suggesting that this activity is strain-dependent and not species-dependent.

Concerning Pep X, the tested strains showed values ranging from 36.55 to 1.18 U/mg (Figure 3). Higher Pep X activity was displayed by the strain Q5C6 (33.22 U/mg), ascribed to the *L. lactis* species, as well as by the *L. delbrueckii* P10 (25.86) strain and by the *L. rhamnosus* P50 (36.55 U/mg) strain. In agreement with our results, some *L. rhamnosus* strains, isolated from semihard artisanal goat cheeses, showed a marked activity for Pep X when tested through Arg-Pro-pNA and Gly-Pro-pNA substrates [27], confirming that *L. rhamnosus* possesses a complex proteolytic system, including the X-prolyl-dipeptidyl aminopeptidase [55]. Differently Psoni et al. (2007) [56], testing *L. lactis* strains isolated from Batzos cheese, revealed Pep X values lower than 5 U/mg. Similarly, Vlieg et al. [57], testing a collection of dairy and wild *L. lactis* strains, obtained values of Pep X activity lower than 1 µmol/mg min. Pep X was purified and characterized from strains ascribed to the *L. acidophilus*, *L. casei*, *L. helveticus*, *L. lactis,* and *S. thermophilus* species [54]. The synergistic effect between Pep N and Pep X is essential to obtain high levels of hydrolysis. In fact, Pep N releases amino acids from the N-terminal of peptides and the rate of hydrolysis is reduced in presence of proline residues [58]. To compensate for the proline inhibition, the Pep X is able to release Xaa-Pro dipeptides from the N-terminal side of peptides. It is advantageous to have strains that can hydrolyze proline-containing peptides, since hydrophobic peptides with at least one proline residue have been linked to the bitter flavor of cheese [59].

## 4. Conclusions

In the present study, a technological characterization of indigenous LAB, isolated from Italian and Brazilian cheeses was carried out. Based on salt resistance, low acidifying and lipolytic activities, ability to produce diacetyl and EPS as well as aminopeptidase activities, data allowed us to select the strains Q5C6, P10, and P50, ascribed to *L. lactis*, *L. delbrueckii,* and *L. rhamnosus*, respectively, as the most promising to be used as adjunct cultures. In particular, the high aminopeptidase N and X activities displayed by the strains could help to enhance the flavor properties of cheese, improving the overall quality of the final product. In addition, the synergistic effect of aminopeptidase N and X revealed by the selected strains could be useful for the reduction of bitter peptides, generated from milk-clotting enzymes during cheese manufacture and ripening.

Research efforts should be made to confirm the results of the present study. The potential flavor improvement of ewe’s and cow’s milk cheese prepared using the selected strains will be further investigated at both pilot and industrial scales.

## Figures and Tables

**Figure 1 foods-12-01154-f001:**
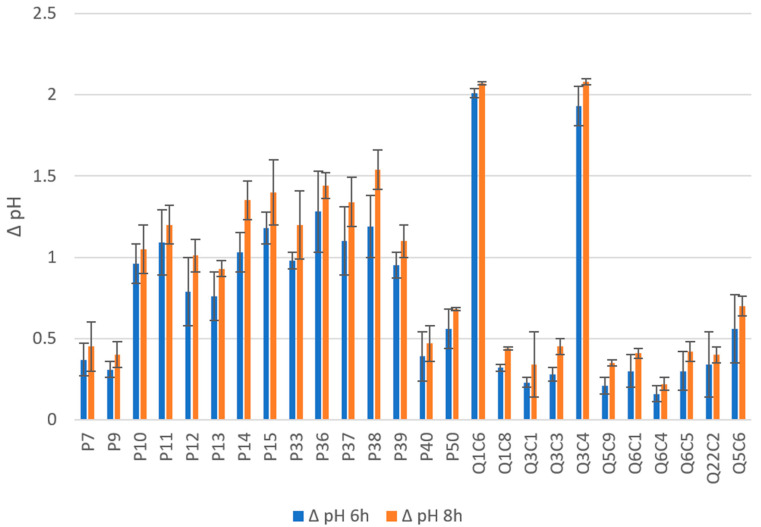
Acidifying activity of LAB strains. Results are expressed as ∆pH (6 h and 8 h) (starting from a pH of 6.7) and reported as mean and standard deviation of three replicates.

**Figure 2 foods-12-01154-f002:**
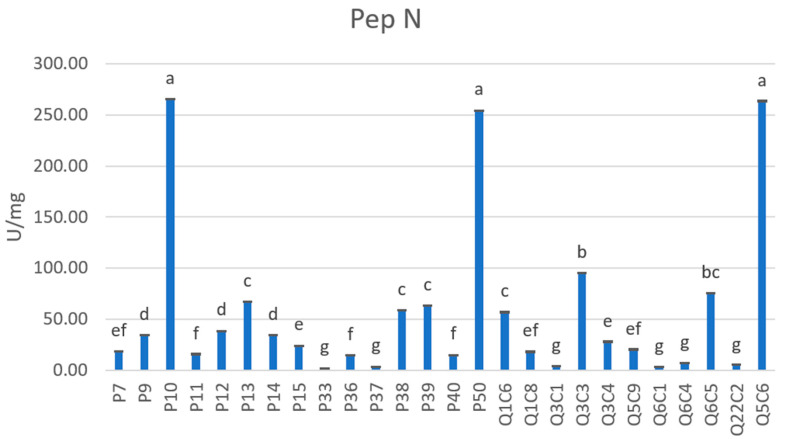
Aminopeptidase N (Pep N) activity exhibited by the tested strains. Results are reported as mean and standard deviation of three replicates. Aminopeptidase activity was expressed as the number of activity units per milligram of protein per minute (U/mg). One unit of aminopeptidase activity was considered as the amount of enzyme required to release 1 μmol of p-NA per minute under the assay conditions. Different letters (a–g) indicate statistically significant differences as determined by the one-way ANOVA test, which is followed by the Tukey’s post-hoc test (*p* < 0.05).

**Figure 3 foods-12-01154-f003:**
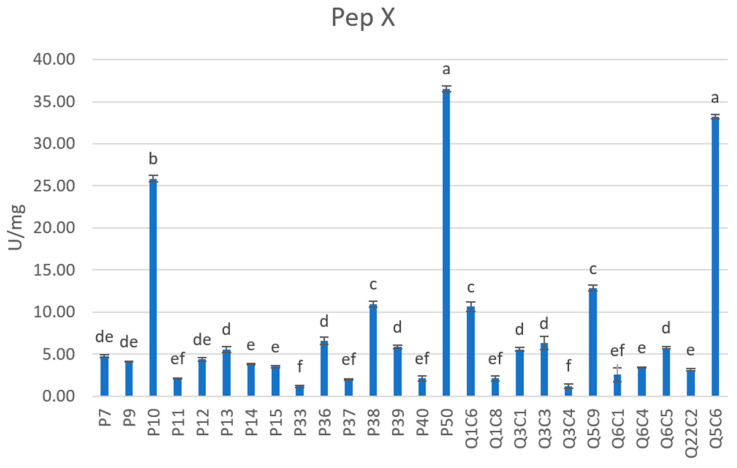
X-prolyl-dipeptidyl aminopeptidase (Pep X) activity exhibited by the tested strains. Results are reported as mean and standard deviation of three replicates. Aminopeptidase activity was expressed as the number of activity units per milligram of protein per minute (U/mg). One unit of aminopeptidase activity was considered as the amount of enzyme required to release 1 μmol of p-NA per minute under the assay conditions. Different letters (a–f) indicate statistically significant differences as determined by the one-way ANOVA test, which is followed by the Tukey’s post-hoc test (*p* < 0.05).

**Table 1 foods-12-01154-t001:** Identification and technological properties of strains isolated from Pecorino cheese, Ragusano cheese, Marajó cheese and Pará cheese.

Isolate	Isolation Source	Species Attribution	Proteolysis *	Lipolysis *	Diacetyl **	EPS *
P7, P9	Pecorino Cheese	*L. delbrueckii*	+	-	++	-
P10	Pecorino Cheese	*L. delbrueckii*	+	-	++	+
P11, P12, P13, P39	Ragusano Cheese	*L. delbrueckii*	+	-	+++	-
P14	Ragusano Cheese	*L. delbrueckii*	+	-	-	+
P15, P33, P40	Ragusano Cheese	*L. delbrueckii*	+	-	-	-
P37	Ragusano Cheese	*L. delbrueckii*	+	-	+	-
P36	Ragusano Cheese	*L. delbrueckii*	+	-	++	-
P38	Ragusano Cheese	*L. delbrueckii*	+	-	+	+
P50	Ragusano Cheese	*L. rhamnosus*	+	-	+	-
Q5C9, Q6C4	Marajó Cheese	*L. plantarum*	+	-	++	-
Q1C6, Q3C4	Marajó Cheese	*L. plantarum*	-	-	+	-
Q1C8, Q6C5, Q22C2	Marajó Cheese	*P. acidilactici*	+	-	+	-
Q3C1	Marajó Cheese	*P. acidilactici*	+	-	+++	-
Q3C3, Q6C1	Marajó Cheese	*P. acidilactici*	+	-	-	-
Q5C6	Pará Cheese	*L. lactis*	+	-	++	-

* (+), positive; (-), negative. ** (+++), strong production; (++), medium production; (+), weak production; (-), no production.

**Table 2 foods-12-01154-t002:** Salt tolerance of LAB strains.

Isolates	2%	6%	10%
P14, P38, P50, Q3C4	+	+	+
P9, P10, P11, P13, P36, P37, P39, Q1C6, Q1C8, Q3C1, Q6C4, Q6C5, Q5C6	+	+	-
P7, P12, P15, P33, P40, Q3C3, Q5C9, Q6C1, Q22C2	+	-	-

(+), positive; (-), negative.

## Data Availability

The data presented is contained within the article.

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
