# Peer review of "Technological Characterization of Lactic Acid Bacteria Strains for Potential Use in Cheese Manufacture"

_foods, 2023, doi:10.3390/foods12061154_

Round 1

Reviewer 1 Report

Dear authors,

            I am ready with critical review of the manuscript- Food_2201510 titled:  Technological characterization of Italian and Brazilian lactic 2 acid bacteria strains for potential use in cheese manufacture.

           The paper present several in vitro tests for enzymes activity with significance for pre-selection of LAB, with potential use in cheese making.

           The title clearly show the objective of the work. I would like to propose a minor edition ex Technological characterization of Italian and Brazilian strains of lactic acid bacteria for potential use in cheese manufacture” The abstract correctly summarized the content and achievements of the study. The paper corresponds well to the aim and scope of the Food journal (MDPI). Overall, the article is well written, in correct and good English without technical errors.

           In my capacity as a reviewer, I would like to ask a few questions:

1.   About the group of 26 LAB strains, mentioned as Italian and Brazilian strains?

·     Please add information about the reason to confer such  specific nationality on the LAB studied.

·     The paper states species affiliation of the LAB strains. Do the species identification was a part of present study or it was published already. If yes, please note these details.

2.   About the selection and presentation of some of the methods for technological evaluation of lactic acid bacteria.

·       the peptidases / dipeptidase activity – I would like to propose to be presented according to the cited method, (ref. 17). Some discrepancy was found between Mat and methods description in the reference  paper and in the present done - method description especially for Units definition done ........’’One unit of aminopeptidase activity was considered as the enzyme amount able to determine an increase in absorbance of 0.001 units.”…page 9, line 337 and 339?

Author Response

Dear authors,

            I am ready with critical review of the manuscript- Food_2201510 titled:  Technological characterization of Italian and Brazilian lactic 2 acid bacteria strains for potential use in cheese manufacture.

           The paper present several in vitro tests for enzymes activity with significance for pre-selection of LAB, with potential use in cheese making.

           The title clearly show the objective of the work. I would like to propose a minor edition ex Technological characterization of Italian and Brazilian strains of lactic acid bacteria for potential use in cheese manufacture” The abstract correctly summarized the content and achievements of the study. The paper corresponds well to the aim and scope of the Food journal (MDPI). Overall, the article is well written, in correct and good English without technical errors.

           In my capacity as a reviewer, I would like to ask a few questions:

  1. About the group of 26 LAB strains, mentioned as Italian and Brazilian strains?
  • Please add information about the reason to confer such  specific nationality on the LAB studied.

Thanks for your suggestion. We agree with you and decided to delete the specification of the nationality of the LAB studied.

  • The paper states species affiliation of the LAB strains. Do the species identification was a part of present study or it was published already. If yes, please note these details.

Thank you for the comment, the species identification was not part of the present study. We added in the text the comment indicating that the work was part of unpublished data.

  1. About the selection and presentation of some of the methods for technological evaluation of lactic acid bacteria.
  • the peptidases / dipeptidase activity – I would like to propose to be presented according to the cited method, (ref. 17). Some discrepancy was found between Mat and methods description in the reference  paper and in the present done – method description especially for Units definition done ........’’One unit of aminopeptidase activity was considered as the enzyme amount able to determine an increase in absorbance of 0.001 units.”…page 9, line 337 and 339?

      Thanks for the comment, we standardized the definition of Unit according to ref. 17. Lines 330-331 and lines 337-338

Reviewer 2 Report

Comments to authors

Journal: Foods

Title:Technological characterization of Italian and Brazilian lactic acid bacteria strains for potential use in cheese manufacture’ the current article focused on the LAB strains and their use in cheese making. The research is interesting, and the manuscript looks good too but still, I have fewer questions as mentioned below.

Comments:

1.     The article have high plagiarism. Need to remove it carefully.

2.     Introduction is short; more data should be added related to the strains and cheese.

3.     The article needs to be revised thoroughly by English experts as it contains many syntax and grammatical errors.

4.     There should be uniformity in the write up for units and symbols in the article.

5.     The authors should describe the importance and need of this research in a paragraph.

Author Response

  1. The article have high plagiarism. Need to remove it carefully.

Thanks for the comment, we removed the plagiarism contained in lines 333-338 and 341-345. However, we have seen that the headers of each page, the affiliations, and some parts of figures 1, 2, and 3 have also been considered plagiarism and that the latter contribute to exceeding 30% of the total. Unfortunately, we can’t change these parts, so we ask you, if necessary, to re-measure the plagiarism content in the text without considering these elements.

  1. Introduction is short; more data should be added related to the strains and cheese.

We added information on traditional Italian and Brazilian Cheeses sources of the tested strains. Lines 66-88.

  1. The article needs to be revised thoroughly by English experts as it contains many syntax and grammatical errors.

Thanks for your comment we revised the manuscript by an English expert.

  1. There should be uniformity in the write up for units and symbols in the article.

      Thank you for the suggestion, we have carefully checked the uniformity of units and symbols throughout the article.

  1. The authors should describe the importance and need of this research in a paragraph.

      Thanks for the comment, the importance of research has been added in the conclusion section.

Reviewer 3 Report

The authors intend to find outstanding or qualified bacterial strains suitable for cheesemaking. It all starts well with appropriate design that seems new to readers, for example, they evaluate the properties of the bacterial strains from several aspects related to the cheesemaking processing, aiming to find one or few good candidates. However, there are several flaws that severely affect the quality of the study. 1. Only few samples (26 samples) have been involved in this study, which may significantly affect the overall results. 2. Parameters for selecting ideal bacterial strains should be expanded and should also closely associate with the property of cheese starter. 3. Usually when one intends to select one good/outstanding candidate, software for statistical analysis such as PCA should be included to optimize the candidates. 4. The manuscript of the study was done in a rush that some uncommon mistakes occur, for example. in line 346,  "be used ad" is a sentence no one understands. 5. The English writing needs to be improved.

Author Response

The authors intend to find outstanding or qualified bacterial strains suitable for cheesemaking. It all starts well with appropriate design that seems new to readers, for example, they evaluate the properties of the bacterial strains from several aspects related to the cheesemaking processing, aiming to find one or few good candidates. However, there are several flaws that severely affect the quality of the study.

  1. Only few samples (26 samples) have been involved in this study, which may significantly affect the overall results.

We agree with the reviewer's comment. Research efforts, including other LAB isolates, will be done.

  1. Parameters for selecting ideal bacterial strains should be expanded and should also closely associate with the property of cheese starter.

Thanks for your comment. The selected strains, as potential adjunct culture candidates, will be screened for additional properties and will be tested in combination with starters, eventually used in cheese manufacture. Like this research studies are ongoing.

  1. Usually when one intends to select one good/outstanding candidate, software for statistical analysis such as PCA should be included to optimize the candidates.

Thanks for your comment. As you suggested, we performed the PCA analysis on aminopeptidase and acidifying activities (since they are the only quantitative data available) and included the plot as a supplementary figure. 

  1. The manuscript of the study was done in a rush that some uncommon mistakes occur, for example. in line 346, "be used ad" is a sentence no one understands.

Thanks for the comment. We revised the manuscript correcting typos.

  1. The English writing needs to be improved.

Thanks for the suggestion, we have improved the writing in English.

Round 2

Reviewer 2 Report

It looks good now.